# Identification of 37 Heterogeneous Drug Candidates for Treatment of COVID-19 via a Rational Transcriptomics-Based Drug Repurposing Approach

**DOI:** 10.3390/ph14020087

**Published:** 2021-01-25

**Authors:** Andrea Gelemanović, Tinka Vidović, Višnja Stepanić, Katarina Trajković

**Affiliations:** 1Mediterranean Institute for Life Sciences (MedILS), Šetalište Ivana Meštrovića 45, 21000 Split, Croatia; tinka.vidovic@medils.hr; 2Ruđer Bošković Institute, Bijenička Cesta 54, 10000 Zagreb, Croatia; visnja.stepanic@irb.hr

**Keywords:** drug repurposing, connectivity map, COVID-19, SARS-CoV-2, host transcriptome

## Abstract

A year after the initial outbreak, the COVID-19 pandemic caused by SARS-CoV-2 virus remains a serious threat to global health, while current treatment options are insufficient to bring major improvements. The aim of this study is to identify repurposable drug candidates with a potential to reverse transcriptomic alterations in the host cells infected by SARS-CoV-2. We have developed a rational computational pipeline to filter publicly available transcriptomic datasets of SARS-CoV-2-infected biosamples based on their responsiveness to the virus, to generate a list of relevant differentially expressed genes, and to identify drug candidates for repurposing using LINCS connectivity map. Pathway enrichment analysis was performed to place the results into biological context. We identified 37 structurally heterogeneous drug candidates and revealed several biological processes as druggable pathways. These pathways include metabolic and biosynthetic processes, cellular developmental processes, immune response and signaling pathways, with steroid metabolic process being targeted by half of the drug candidates. The pipeline developed in this study integrates biological knowledge with rational study design and can be adapted for future more comprehensive studies. Our findings support further investigations of some drugs currently in clinical trials, such as itraconazole and imatinib, and suggest 31 previously unexplored drugs as treatment options for COVID-19.

## 1. Introduction

Coronavirus disease 2019 (COVID-19) is a new infectious disease caused by the severe acute respiratory syndrome coronavirus 2 (SARS-CoV-2). The common clinical features of the COVID-19 range from mild respiratory symptoms to severe acute respiratory distress syndrome, and may be accompanied with a wide spectrum of gastrointestinal, cardiovascular, or neurological manifestations [1]. COVID-19 first appeared in December 2019 in Wuhan, China, where the first pneumonia cases of unknown origin were reported [2]. The disease has spread rapidly around the globe and the World Health Organization (WHO) declared a COVID-19 pandemic on March 11th 2020 [3]. By 23 December 2020 the number of infected subjects has reached over 78 million with more than 1.7 million deaths worldwide. Currently, there are over 21 million active cases, with about 0.5% of patients in critical condition [4,5]. Overall, COVID-19 has caused numerous socioeconomic consequences in every aspect of daily life, ranging from great economic loses in all sectors, challenges in the healthcare system, travel restrictions, social distancing and lockdowns [6], all of which have seriously impacted mental health, with higher rates of anxiety, depression and stress reported among the general population [7].

To date, several vaccines have been developed to prevent further spreading of COVID-19 [8]. Furthermore, massive efforts have been undertaken to find effective treatments for those who had already contracted the disease—at present, there are more than 4000 clinical studies related to COVID-19 as listed on ClinicalTrials.gov database, where majority were focused on chloroquine or hydroxychloroquine with or without azithromycin, lopinavir/ritonavir, remdesivir or dexamethasone [9]. Except for dexamethasone which led to improvement in survival of hospitalized patients in need for supplemental oxygen [10], none of the aforementioned drugs showed significant efficacy in ameliorating the COVID-19 outcome [11,12,13]. Some of the other identified drugs, such as ivermectin, ibrutinib, imatinib or ruxolitinib, are currently in different phases of clinical trials, but so far none of them has been recommended for treatment of COVID-19 by the COVID-19 Treatment Guidelines Panel [14]. Only two drugs are currently approved for COVID-19: in October 2020 U.S. Food and Drug Administration (FDA) approved remdesivir as the first drug for the treatment of severe COVID-19 cases in need for hospitalization [15], and in November 2020 the same entity issued an emergency use authorization for the drug baricitinib [16]. However, since these drugs seem to have limited efficiency and serious side effects [17], it is urgent to identify more potent and safe therapeutics that could significantly decrease mortality and relieve the burden of COVID-19 on human health and healthcare systems worldwide. 

In conventional circumstances, the process of developing new drugs and testing their safety and efficacy in clinical trials takes several years, and the median cost of this process is almost a billion USD per drug [18]. The fastest and cheapest way to discover therapeutics for COVID-19 is repurposing of already approved drugs. Multiple experimental and computational drug repurposing strategies have been developed to discover novel indications for well characterized drugs that had already passed extensive clinical studies [19]. Examples of successful drug repurposing include sildenafil—originally an antihypertensive drug repurposed for erectile dysfunction, thalidomide—sedative that is also effective against erythema nodosum leprosum and multiple myeloma, or aspirin—an analgesic that can also be used to decrease the risk of cardiovascular disease and colorectal cancer [19].

In the past decade, a principle of transcriptomic signature reversion has been increasingly employed as a computational drug repurposing strategy, especially in cancer research [20]. This approach identifies drugs with inverted transcriptomic signatures in relation to the signature of a disease. Treatment of patients with such drugs could thus potentially reverse the disease transcriptomic signature, presumably ameliorating the disease phenotype as a result [19]. Signature matching of transcriptomic data started in 2006 with The Connectivity Map (CMap) project [21] which further evolved into the Library of Integrated Network-Based Cellular Signatures (LINCS) [22,23]. Transcriptomic signature reversion approach using CMap has been extensively used in pharmacogenomics studies. Examples of experimentally validated drugs that were identified by this approach include antiepileptic drug topiramate that showed potential for treating inflammatory bowel disease [24], mTOR inhibitor rapamycin shown to induce glucocorticoid sensitivity in acute lymphoblastic leukemia [25], and ikarugamycin and quercetin shown to reduce inflammation in cystic fibrosis [26]. Even though clinical trials are still missing to provide a definite proof-of-concept, such approach has demonstrated a potential for drug prioritization in pharmacogenomics research [20].

Several in silico drug repurposing studies based on a transcriptome reversal approach have already been published in the context of COVID-19 [27,28,29,30,31]. However, one common issue in such studies is a complete lack or insufficiency of the criteria for inclusion of datasets in the analysis which may then produce misleading results. Careful consideration of biological parameters, such as, for example, host cell tropism of the virus, is needed to assess the suitability of each dataset. Furthermore, high variability among datasets and noise in the transcriptomes often cast doubts on the validity of the results [32]. Combining biological knowledge with bioinformatics approaches is much needed to ensure the validity of such studies and to increase chances that selected drugs will be efficient in suppressing disease symptoms. 

In this study, we used the CMap computational drug repurposing approach to identify drug candidates with the potential to revert host transcriptome alterations triggered by the SARS-CoV-2 virus. Specifically, we searched publicly available transcriptomic datasets deposited via Gene Expression Omnibus (GEO) [33] of SARS-CoV-2 infected cells with matching non-infected controls, identified differentially expressed genes (DEGs), and employed LINCS [22] database to select drugs with transcriptomic signatures opposite to those induced by the SARS-CoV-2 infection. Of note, we used a rational strategy based on unsupervised machine learning approaches to select biologically relevant datasets, i.e., those that exhibited relatively high magnitude of response to viral infection. Moreover, we filtered DEGs to obtain those that are shared among multiple datasets so that they reflect a general and robust response to infection which should correspond to a physiological state of an infected organism. The obtained final list of 37 drug candidates was then characterized using bio- and chemoinformatic analyses to provide additional insights into the pathogenesis and viral-host interaction mechanism(s).

## 2. Results

In this study we determined transcriptome alterations in cells infected with SARS-CoV-2 in relation to non-infected controls and used these data to reveal drugs that induce the opposite-sense changes in the subset of the transcriptome affected by the virus. We propose that these drugs could reverse the host transcriptomic signature induced by SARS-CoV-2 and are thus potential candidates for treatment of COVID-19. Overall study design is depicted in Figure 1.

### 2.1. Selection of the Relevant Datasets

At first, publicly available transcriptomes obtained from cells infected with SARS-CoV-2 and their non-infected counterparts were identified. Considering heterogeneity in design of the identified studies, datasets for the analysis were pre-selected based on the pre-defined criteria (see Materials and Methods). As a result, the seven transcriptomic datasets were included in our analysis (Table 1, Appendix A). These datasets were obtained from four cell lines: normal human bronchial epithelial cell line (NHBE), human lung adenocarcinoma alveolar epithelial cell line (A549) with and without overexpressed angiotensin-converting enzyme 2 (ACE2; host receptor to which SARS-CoV-2 binds and enters the cell), and human lung adenocarcinoma airway epithelial cell line (Calu-3). For three cell lines datasets obtained upon infection with two different multiplicity of infection (MOI) were included in the analysis (Table 1). To provide an additional source of data for NHBE, we included a dataset obtained from human bronchial organoids generated from NHBE cell line, even though this dataset did not fully match the inclusion criteria (the samples were collected for RNA-seq analysis 5 days post-infection instead of 24 h post-infection).

We next identified DEGs in each infected sample relative to corresponding non-infected control (Appendix A, Appendix A). Interestingly, we observed high variation in the number of DEGs among different datasets, indicating that some samples are more responsive to SARS-CoV-2 than the others. We thus decided to reduce the list of analyzed datasets to only those with relatively high magnitude of response to the virus in terms of transcriptome changes. To that end, we performed PCA on all eight transcriptome datasets and found that the distance on the score plot between infected and non-infected counterparts was apparently larger for A549-ACE2 and Calu-3 (regardless of the MOI) as compared to A549, hBO and nHBE samples (Figure 2). This PCA analysis indicated that the magnitude of response to the virus was higher for A549-ACE2 and Calu-3 relative to A549 without ACE2 overexpression, hBO and nHBE samples.

To additionally evaluate the effect of MOI, the separate PCA analyses were conducted for each cell line infected with two MOIs (Appendix A). In A549-ACE2 and Calu-3 cell lines, there were more differences between non-infected and infected samples, than between samples treated with different MOI, indicating high sensitivity of these lines to low quantities of the virus. Conversely, A549 cells showed negligible changes in the transcriptome upon infection with low MOI of 0.2 as compared to high MOI of 2, suggesting high threshold for infection. Next, differential gene expression analyses were performed for A549, A549-ACE2, and Calu-3 cell lines, considering pairs of SARS-CoV-2 infected and non-infected samples with different MOI as covariate (Appendix A, Appendix A). Combining transcriptomes of different MOIs for each cell line in one analysis would correspond to a general and robust response to SARS-CoV-2 infection which much better represents physiological situation where different cells are exposed to a range of quantities of viral particles. Furthermore, pooling two different MOIs in this kind of analysis contributes to reduction of noise in the data. In order to reduce the noise in the dataset obtained from NHBE cells for which there were no available transcriptomes upon infection with different MOIs, we performed an analogous analysis of NHBE cells pooled with hBOs (DEG analysis and PCA), given that hBOs were generated from NHBE cells (Appendix A, Appendix A).

To create final selection of datasets for further analysis which takes into account the magnitude of response to the virus and the effect of different MOI, we performed hierarchical clustering on the results of the four differential gene expression analyses (Appendix A). This analysis yielded two distinct clusters: one that contained A549-ACE2 and Calu-3 infected with SARS-CoV-2 at both MOIs, and another that included A549, hBO and NHBE samples. Hence, we opted to continue our study only on A549-ACE2 and Calu-3 cell lines, whereby the data for both MOIs were included.

### 2.2. Selection of the Relevant DEGs upon SARS-CoV-2 Infection

The next objective was to identify the most relevant DEGs that reflect robust change in the host transcriptomic signature upon SARS-CoV-2 infection in more than one cell type (Appendix A). To that end, we overlapped DEGs obtained from the two selected differential gene expression analyses (A549-ACE2 and Calu-3 infected with low and high MOI) and selected only the DEGs that were shared among these datasets. Furthermore, only DEGs whose expression consistently changed in the same direction—either up or down were considered. This procedure resulted in the list of 636 DEGs (Appendix A).

The cellular pathways affected by SARS-CoV-2 infection were determined by performing Gene Ontology (GO) enrichment analysis of these 636 DEGs. The analysis showed that SARS-CoV-2 infection upregulates various cellular responses such as immune and neuroinflammatory response pathways, and downregulates cold-induced thermogenesis and cholesterol biosynthetic pathways (Figure 3, Appendix A). As expected, one of the GO categories affected by SARS-CoV-2 infection was “host defense against viral infection”. Since host defense against viral infection represents a beneficial mechanism for the cells and presumably should not be reverted by drugs, we excluded a total of 97 overexpressed genes that fell into this GO category (Appendix A). The resulting 539 DEGs were employed in the subsequent steps of our study (Appendix A, Appendix A).

### 2.3. Identification of Drugs with a Potential to Reverse Transcriptomic Signature Upon SARS-CoV-2 Infection

To identify drugs capable of inducing inverted transcriptomic signature in the host relative to transcriptional changes triggered by SARS-CoV-2, we performed a LINCS connectivity map analysis of the final list of DEGs upon SARS-CoV-2 infection (Appendix A). Although it included data obtained in as many as 30 cell lines, at least for some compounds, LINCS does not provide data specifically for the selected A549-ACE2 and Calu-3 cells. Furthermore, some drugs display highly variable effects in different cell lines making it difficult to predict how they would affect the two lines of interest. To increase probability that the selected drugs would have analogous effects on A549-ACE2 and Calu-3 cell lines, we filtered out compounds with documented highly variable effects across multiple cell lines. This additional filtering step led to the retention of drugs with robust effects across multiple cell lines. In addition, we focused on already approved drugs with the aim of their repurposing (see Materials and Methods). The final list includes 37 drug candidates with a potential to reverse transcriptomic signature upon SARS-CoV-2 infection (Table 2). These drugs meet the following criteria: (1) have a transcriptional signature opposite of the signature of the SARS-CoV-2 infection; (2) have significant effects in single cell lines or significant and robust effects in multiple cell lines; (3) are already approved for human use. Of note, some drugs showed very high connectivity score, meaning that they are particularly potent in reversing transcriptomic signature in at least one cell line, and they include antineoplastic drug imatinib (Tau −99.33), neuroprotective memantine (Tau −97.66), antiarrhythmic ibutilide (Tau −96.53), antibacterial azithromycin (Tau −96.18), anticonvulsant trimethadione (Tau −96.08) and antifungal itraconazole (Tau −95.47). On the other hand, some drugs had a very robust effect across multiple cell lines, which makes them more likely to be efficient in the two lines of interest. The latter subset of drugs includes calcimimetic drug cinacalcet and antipsychotic drug fluspirilene that were efficient in 6 cell lines and antifungal drug itraconazole efficient in 5 cell lines.

### 2.4. Bio- and Chemoinformatic Characterization of the Drug Candidates for Repurposing against SARS-CoV-2 Infection

To evaluate whether 37 drug candidates share some biological and molecular properties, we performed clustering of the drugs based on the following parameters: pharmacological class and current indication, mechanism of action (MOA), molecular structure, and known protein targets (Appendix A). In terms of their current indication, we found that 11 of the drugs cluster as antiinfective agents, 7 as neuropsychiatric drugs, and 5 as cardiovascular drugs, whereas the remaining drugs are pharmacologically heterogeneous (Appendix A). The MOAs of these 37 drugs were also heterogeneous (Appendix A). The clusters with two or maximum three drugs were bacterial ribosomal subunit inhibitors, bacterial topoisomerase II inhibitors, 14-alpha demethylase inhibitors, histamine receptor antagonists, and dopamine receptor antagonists. Furthermore, no significant similarity of the molecular structures was found among the drugs, with the maximum Tanimoto coefficient of 0.54 obtained for itraconazole and ketoconazole, which is still lower than the threshold of 0.85 above which drugs are considered significantly similar (Appendix A). Selected drug candidates are also heterogeneous based on their physicochemical properties (Appendix A). Grouping of the drugs based on their protein targets reveals no obvious preference for any of the four main drug target groups: G-protein coupled receptors (GPCR), ion channels, kinases, nuclear receptors (Appendix A). Indeed, majority of the selected drugs target GPCRs (14/37, 38%), and most drug targets are membrane-bound (174/282, 62%; Appendix A), as in the case when the unfiltered list of all existing drugs is considered [37,38,39]. In summary, this analysis illustrates that the 37 drugs with a potential to reverse SARS-CoV-2 transcriptomic signature are highly heterogeneous in terms of their properties, with main clusters based on their current therapeutic indication (Appendix A).

In order to find out which specific biological pathways affected by SARS-CoV-2 infection can be reversed by the selected 37 drug candidates, we performed Drug and Target Set Enrichment Analysis (DSEA and TSEA). Biological pathways regulated by these drugs (Appendix A) were overlapped with the pathways affected by the virus identified in the previous step of this study (Figure 3, Appendix A). The overlap contains the following categories: metabolic and biosynthetic process, immune system process, cellular and tissue developmental process, cellular architecture and dynamics, signaling pathways, and response to stimulus, with steroid metabolic process being targeted by almost half of the selected drug candidates (Figure 4, Appendix A). We conclude that these pathways, in particular the steroid metabolic processes, might be the key druggable pathways for the reversal of SARS-CoV-2 transcriptomic signature.

## 3. Discussion

In this study we used a rational approach to filter currently available transcriptomic datasets and determine relevant DEGs upon SARS-CoV-2 infection, followed by using this information to identify drug candidates with inverse transcriptomic signature as compared to SARS-CoV-2-induced transcriptome changes. This work revealed 37 diverse drug candidates that could potentially reverse SARS-CoV-2 signature through targeting a range of biological pathways, including immune response, metabolic and biosynthetic process, cell differentiation and proliferation, and signaling pathways.

To increase chances that the obtained drug candidates for repurposing will be efficient in vivo, we applied multiple measures of caution during development of our bioinformatics pipeline. This was achieved by introducing several filtering steps on each level of the analysis and these include: (1) selection of transcriptomic datasets obtained from biosamples whose transcriptomes are highly responsive to SARS-Cov-2 infection; (2) reduction of noise in the available transcriptomic data; (3) removal of the genes important for cellular defense against virus from the list of target DEGs, and (4) removal of the drugs with documented variable effects across different cell lines. At the time when this study was performed several datasets obtained from various cells or tissues were available. However, not all tissues or cells are equally affected by SARS-CoV-2 and hence not all datasets have equal value as a source of transcriptomic data. Indeed, the host cell tropism of SARS-CoV-2 depends on the cellular expression of factors that control viral entry and reproduction. For instance, to successfully infect the host cells, SARS-CoV-2 requires the presence of angiotensin-converting enzyme 2 (ACE2) and transmembrane serine protease 2 (TMPRSS2) at the cellular surface [40]. Moreover, different cell types vary in their ability to support production of the new virions [41]. In line with these findings, we observed dramatic differences in transcriptome alterations among several cell types exposed to SARS-CoV-2. Using PCA and hierarchical clustering on the selected cell types (Figure 2, Appendix A) based on their total transcriptomes and their DEGs, we were able to select the two cell lines—A549 cells expressing ACE2 and Calu-3 cells for further analyses. These cell lines were relatively sensitive to the virus, i.e., their transcriptomes were more responsive to viral infection as compared to wild-type A549 cells, NHBE cells and hBO. The selected cell types are presumably more vulnerable to the infection and thus represent priority targets for therapeutic intervention. Of note, our results are in agreement with the previously published data suggesting that A549 and NHBE have low or variable levels of ACE2 receptor [34,42,43] and are thus not ideal models for studying SARS-CoV-2 infection.

Data noise is a common issue in the analysis of transcriptomes obtained from different sources or from limited number of samples. Correspondingly, we have observed high variability in DEGs among different datasets. To reduce the noise, we harmonized the transcriptomic data by including only datasets that met a set of defined criteria and by filtering datasets as described above. Finally, only consistent DEGs—those that were shared between the two selected biosamples, were used in further analyses.

Upon viral infection, many biological pathways are hijacked by the virus and used for production of new virions. However, cellular defense against the virus is also activated. Reversal of the expression of genes that belong to the latter pathway would obviously represent a disadvantage for the cell, at least in the early stages of infection. Therefore, we performed a gene ontology enrichment to classify all affected DEGs into various biological pathways. To ensure that the selected drugs will not affect cellular defense against virus, we omitted all genes that fell into this category from the list of DEGs used for drug selection.

To identify drugs that could be repurposed for COVID-19 we used the filtered subset of DEGs as an input for LINCS connectivity map analysis. Since LINCS does not contain data specifically for the selected cell lines, and many drugs display cell type-specific effects, an additional parameter was introduced in the analysis – the robustness of drug effects across multiple cell types. In that we eliminated drugs for which we found evidence that their effects are not conserved across multiple cell lines. This step is important as it increases the likelihood that the selected drugs will have the desired effect in other cell types, including our biosamples of interest.

The above described pipeline (Figure 1) resulted in the list of 37 drug candidates with potential to reverse SARS-CoV-2 transcriptomic signature. However, bio- and chemoinformatic analysis of these drugs revealed that they are diverse in terms of their chemical structure, physicochemical properties, targets and biological pathways that they affect, which is in agreement with our observation that DEGs upon SARS-CoV-2 infection are involved in multiple cellular processes. Therefore, it is possible that each small cluster of drugs could reverse a different subset of SARS-CoV-2 transcriptome signature. It is tempting to speculate that a combination of two or more individual drugs belonging to different clusters could have more potent transcriptome reversal effects as compared to using a single drug.

We identified several biological processes affected by SARS-CoV-2 that can be targeted by drugs, and these include metabolic, developmental, immune, and signaling processes (Figure 4). Interestingly, steroid metabolic process was at the top of these processes as it was targeted by half of the selected drugs. This result is in agreement with a documented role of cholesterol in the infection of the cells by another coronavirus, transmissible gastroenteritis virus [44], as well as by a porcine nidovirus [45]. Furthermore, cholesterol is an important constituent of the cellular membranes, and those are essential for almost all aspects of the viral life cycle, including the attachment of the virus to the cell surface, fusion of the virus with the plasma membrane and/or endosomes, viral replication in double-membrane vesicles and budding of the virus from intracellular membrane compartments [46]. Finally, steroids have a substantial effect on host immune response [47,48,49].

Some of the drugs from our list are already being tested for their effects against COVID-19. These include antiinfective drugs ritonavir, azithromycin, atovaquone and itraconazole, antineoplastic drug imatinib and antidepressant drug fluoxetine [9]. Furthermore, azithromycin, one of the drug candidates selected in this study, was previously identified by a recently published network-based approach [50]. Also, at least three receptors that are targeted by drugs identified in this study were also suggested as putative targets in other network-based approaches. These include sigma non-opioid receptor 1 (SIGMAR1) [51,52] targeted by nortriptyline, and beta-2 adrenergic receptor (ADRB2) and androgen receptor (AR) [53] targeted by nortriptyline, levobunolol and ketoconazole. Our results lend support for further investigation of these drugs or drug targets in experimental approaches to treatment of COVID-19. Finally, this study suggests novel drug candidates for COVID-19 treatment, such as memantine, ibutilide, or trimethadione.

In comparison with other studies which employed similar computational approach for drug repurposing based on transcriptome reversal [27,28,29,31], we observed only a minor or no overlap of the drug candidate lists. Shared candidates were ADRA1B antagonists (nortriptyline in our study), as well as ACE inhibitor perindopril and NR1I2 agonists (econazole and ritonavir in our study) that were also identified by El-Hachem et al. [30]. This limited agreement between similar studies may stem from using different starting transcriptomic datasets as well as from differences in the criteria applied in the dataset and DEGs selection procedure.

The main strengths of this study are a sound study design and integration of current biological knowledge with rigorous statistics. The pipeline we developed employs a rational and biologically relevant selection of the datasets and differentially expressed genes with the aim to increase reliability of the results. A limitation of this work is that it has been performed in the early phase of COVID-19 investigations on a relatively small number of available datasets. Moreover, the selected datasets were obtained from cancer cell lines which are not an optimal source of transcriptomic data for studying a cancer-unrelated disease such as COVID-19. Nevertheless, our pipeline could be applied in future more comprehensive studies upon publication of transcriptomic datasets obtained from more relevant biosamples such as SARS-CoV-2-infected primary human cell lines. This approach would also benefit from more profound understanding of the cellular tropism of SARS-CoV-2 and of vulnerability of different primary cell lines to the virus. In such future study, our pipeline could be refined to incorporate an additional filtering step of the datasets with positive selection of the vulnerable cell lines and negative selection of the indifferent cell lines, whereas drugs could be additionally filtered based on their selective efficiency only in the relevant cell types to avoid bystander toxicity. The pipeline could also be further upgraded to address more complex questions such as temporal dimension of transcriptome changes upon SARS-CoV-2 infection and timing of drug treatments. This would require more knowledge on the dynamic nature of cellular changes post-infection. Larger number of the relevant transcriptomic datasets could then be analysed after their initial clustering based on time post-infection and cell type. Finally, while in this study we focused only on robustly affected genes with fold change higher than 2 due to the limited number of datasets, the developed pipeline could also be further refined by optimizing fold change threshold in a gene-specific manner, given that fold change in expression does not have equal biological effects for all genes. This will also be possible upon generation of more transcriptomic datasets and of more profound knowledge about genes and pathways that are key to the pathogenesis of COVID-19.

## 4. Materials and Methods

### 4.1. Publicly Available Transcriptomics Datasets

Publicly available transcriptomic datasets were identified by the extensive search of Gene Expression Omnibus (GEO) repository (https://www.ncbi.nlm.nih.gov/gds/) [33] using keywords “SARS-CoV-2 OR COVID-19” (initial screening was performed on 7 July 2020 with a follow-up on 20 October 2020). Datasets for the analysis were pre-selected based on the following criteria: (1) human samples; (2) transcriptional profile available for non-infected (mock-treated) and infected with SARS-CoV-2 samples; (3) only in vitro studies on cells or organoids of bronchial/lung-origin; (4) minimum of two biological replicates; (5) only bulk RNA-seq technology; (6) cell samples harvested for RNA-seq 24 h post-infection. Raw counts from the selected transcriptomic datasets were downloaded from GEO repository.

### 4.2. Differential Gene Expression Analyses

The complete bioinformatics pipeline was performed in the free software environment for statistical computing R, version 4.0.0 [54]. Differential gene expression analysis was performed with the R package *DESeq2* version 1.28.1 [55]. Raw counts from each of the included transcriptomic datasets were first pre-filtered to remove genes with read counts lower than 10. The remaining raw counts were normalized using DESeq2 variance stabilizing transformation (VST). PCA analysis was performed on the normalized raw counts. For further downstream analysis only DEGs with false discovery rate (FDR) adjusted *p*-value < 0.05 and fold change >2 for upregulated genes or <0.5 for downregulated genes were considered. Hierarchical clustering of datasets was performed with DEGs as an input with Euclidean distance measure and complete linkage as a clustering method, using base R function *hclust*. R package *BiomaRt* version 2.44.0 [56,57] with Ensembl database was used to convert gene names to Entrez ID for downstream analysis. Functional enrichment analysis was performed with the R package *clusterProfiler* version 3.16.0 [58]. GO over-representation test was done separately for up- and downregulated DEGs and the results were filtered based on FDR adjusted *p*-value less than 0.05. Redundant GO terms were removed by applying semantic similarity method implemented within the function *simplify*, using the similarity cut-off of 0.4 [59].

### 4.3. Library of Integrated Network-Based Cellular Signatures (LINCS) Database Analysis

Transcriptomic signatures induced by SARS-CoV-2 infection were compared with the signatures induced by treatments with various small molecule compounds using the CMap analysis approach. The CMap analysis was conducted using LINCS reference database Phase 1 via an R package *signatureSearch* version 1.2.5 [60]. Within *signatureSearch*, LINCS reference database consisted of differential gene expression analysis of 12,328 genes obtained upon treatments of 30 cell lines with 8140 compounds as perturbagens, which corresponded to a total of 45,956 signatures [60]. The results of LINCS analysis are lists of perturbagen-cell line connectivity scores represented by Tau (Tau is a standardized score ranging from −100 to 100, where more negative/positive value signifies more extensive reversal/enhancement of transcriptomic signature by a perturbagen in a given cell line) [22].

The obtained list of signatures was further filtered according to the following pipeline:(1)FDR adjusted *p*-value of weighted connectivity score was given for each perturbagen-cell line combination. Only significant combinations with FDR adjusted *p*-value less than 0.05 were selected.(2)Tau connectivity score was given for all significant perturbagen-cell line combinations. Wherever a perturbagen was tested in multiple cell lines, the mean Tau connectivity score and its coefficient of variation (CV, described as the standard deviation divided by the mean) were calculated. Only perturbagens with CV < 1, i.e., those that showed coherent transcriptomic signature in multiple cell lines were chosen. Finally, all perturbagens with Tau < −85 were filtered for further analysis. The recommended Tau threshold of −90 was lowered to −85 to increase the final number of identified drug candidates.(3)The list of perturbagens was additionally reduced to include only approved drugs which were used for downstream analysis. Information about drug approval status was obtained via CLUE Repurposing App (https://clue.io/repurposing-app/; selection of 2427 drugs in launched phase).

### 4.4. Bio- and Chemoinformatic Analyses of Candidate Drugs

Information about drugs (molecular formula, molecular structure (as canonical Simplified Molecular Input Line Entry System (SMILES)), chemical class, pharmacological class, current indication based on The Anatomical Therapeutic Chemical (ATC) classification, mechanism of action (MOA) and cellular location) was collected from PubChem online database (https://pubchem.ncbi.nlm.nih.gov/) [61]. Physicochemical profiles of the drugs were estimated using ADMET Predictor^TM^ 9.5 software (Simulations Plus, Inc., USA) with canonical SMILES of compounds as inputs [62]. Parameter relative polar surface area (RelPSA) was calculated using DataWarrior software [63]. Ionisation states of the drugs were estimated from acidity and basicity ionization constants calculated by ADMET Predictor^TM^ 9.5 software. Information on drug target and the type of drug-target interaction was obtained from online databases DrugBank (https://go.drugbank.com/) [64] and Drug Gene Interaction Database (DGIdb; https://www.dgidb.org/) [65]. Cellular location of drug targets was extracted from DrugBank. Information about drug target protein families and superfamilies was obtained from UniProtKB (https://www.uniprot.org/) [66] and InterPro (https://www.ebi.ac.uk/interpro/) [67], while information on enzyme class was obtained from Integrated relational Enzyme database (IntEnz; https://www.ebi.ac.uk/intenz/) [68]. Functional enrichments on the levels of drugs (Drug Set Enrichment Analysis (DSEA)) and targets (Target Set Enrichment Analysis (TSEA)) were performed with *signatureSearch* using hypergeometric test function and GO annotation. Results were filtered based on FDR adjusted *p*-value less than 0.05 and redundant GO terms were removed using REVIGO online tool (http://revigo.irb.hr/) [69] with similarity cut-off of 0.7.

Clustering of the drugs was performed in the following steps: (1) for structural similarity only, canonical SMILES were transposed into circular ECFP6 (extended-connectivity fingerprint of diameter 6) fingerprints using R package *rcdk* version 3.5.0 [70] with default options; (2) similarity matrix was calculated from binary (or ECFP6 in case of structural similarity) fingerprints with default Tanimoto similarity metric using package fingerprint version 3.5.7 [71]; (3) hierarchical clustering was performed using base R function *hclust* with distance matrix as input (1 – Tanimoto similarity metric) and default option of complete linkage as a clustering method.

### 4.5. Preparation of Figures

All figures (except pipelines and drug-target-pathway network) were designed in R, version 4.0.0 [54] using the following packages: *ggplot2* version 3.3.2 to visualize results of PCA analysis and create barplots [72], *dendextend* version 1.14.0 to visualize results of hierarchical clustering as dendrogram [73], and *clusterProfiler* version 3.16.0 for depicting results of GO enrichment analysis [58]. Drug-target-pathway network was visualized using open source software for network visualization Cytoscape version 3.7.1 [74].

## Figures and Tables

**Figure 1 pharmaceuticals-14-00087-f001:**
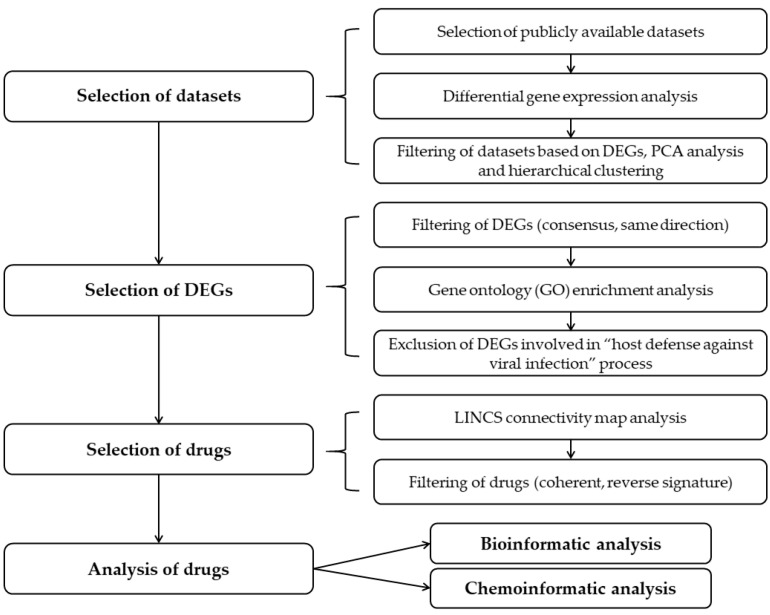
Overall study design for identification of candidate drugs that could reverse transcriptomic signature upon SARS-CoV-2 infection.

**Figure 2 pharmaceuticals-14-00087-f002:**
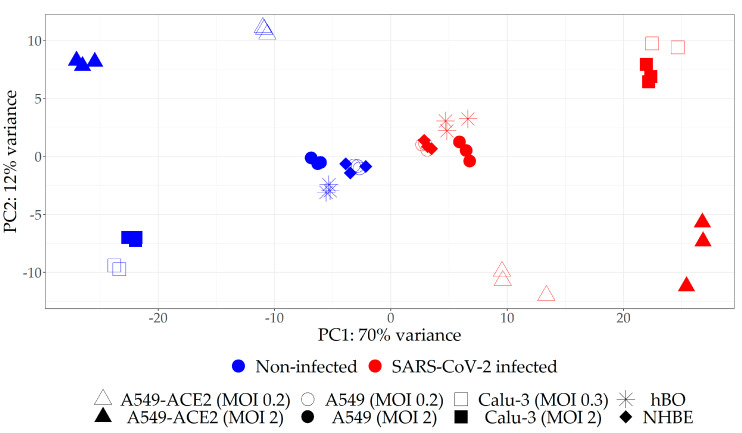
The PCA score plot of all included transcriptomic datasets indicating that A549-ACE2 and Calu-3 cells were more responsive to SARS-CoV-2 infection relative to A549 cell line without ACE2 overexpression, hBO as well as NHBE cells.

**Figure 3 pharmaceuticals-14-00087-f003:**
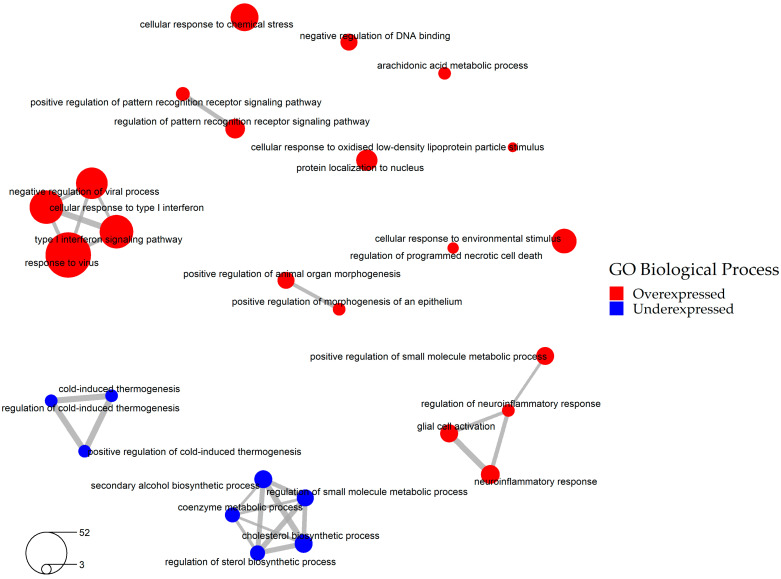
Network of significantly enriched pathways involved in SARS-CoV-2 infection (Gene Ontology Biological Process database, based on 636 consensus DEGs).

**Figure 4 pharmaceuticals-14-00087-f004:**
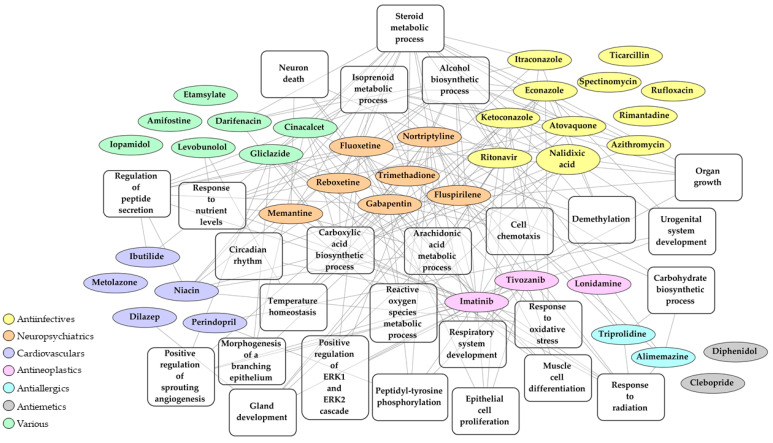
Network of 37 repurposable drug candidates that may reverse SARS-CoV-2 transcriptomic signature based on current therapeutic indication and key biological pathways.

**Table 1 pharmaceuticals-14-00087-t001:** Publicly available transcriptomic datasets used in the study.

Study Label	Description	No. Samples (H/I *)	GEO Accession	Reference
**Cells collected for RNA-seq 24 h post-infection**				
A549 (MOI 0.2)	Human lung adenocarcinoma alveolar epithelial cells	3/3	GSE147507	[34]
A549 (MOI 2)	Human lung adenocarcinoma alveolar epithelial cells	3/3		
A549-ACE2 (MOI 0.2)	Human lung adenocarcinoma alveolar epithelial cells with overexpressed ACE2	3/3		
A549-ACE2 (MOI 2)	Human lung adenocarcinoma alveolar epithelial cells with overexpressed ACE2	3/3		
Calu-3 (MOI 2)	Human lung adenocarcinoma airway epithelial cells	3/3		
NHBE (MOI 2)	Normal human bronchial epithelial cells	3/3		
Calu-3 (MOI 0.3)	Human lung adenocarcinoma airway epithelial cells	2/2	GSE148729	[35]
**Organoids collected for RNA-seq 5 days post-infection**				
hBO	Human bronchial organoids generated from NHBE cells	3/2	GSE150819	[36]

* Non-infected sample/SARS-CoV-2 infected sample; MOI: multiplicity of infection; GEO: Gene Expression Omnibus database.

**Table 2 pharmaceuticals-14-00087-t002:** Final list of 37 repurposable drug candidates with a potential to reverse transcriptomic signature upon SARS-CoV-2 infection (sorted in alphabetical order).

Drug	Mean Tau	N Cell Lines with Same Effect	Pharmacological Class (Current Indication)	Mechanism of Action (MOA)
Alimemazine	−89.98	4	Antiallergic agent	Histamine receptor antagonist
Amifostine	−90.18	1	Radiation protective agent	Free radical scavenging activity
Atovaquone	−89.15	4	Antiinfective agent (antiprotozoal)	Protozoal mitochondrial electron transport inhibitor
Azithromycin	−96.18	1	Antiinfective agent (antibacterial)	Bacterial 50S ribosomal subunit inhibitor
Cinacalcet	−88.68	6	Calcimimetic agent	Calcium-sensing receptor agonist
Clebopride	−93.96	1	Antiemetic agent	Dopamine receptor antagonist
Darifenacin	−91.02	1	Anticholinergic agent	Cholinergic muscarinic antagonist
Dilazep	−88.20	4	Antihypertensive agent (vasodilatator)	Adenosine reuptake inhibitor
Diphenidol	−88.98	1	Antiemetic agent	Acetylcholine receptor inhibitor
Econazole	−91.43	1	Antiinfective agent (antifungal)	Fungal cytochrome P450 inhibitor (14-alpha demethylase inhibitors)
Etamsylate	−91.82	1	Hemostatic agent	Hemostatic
Fluoxetine	−92.79	2	Antidepressant agent	Selective serotonin reuptake inhibitor
Fluspirilene	−93.58	6	Antipsychotic agent	Dopamine receptor antagonists
Gabapentin	−86.68	2	Anticonvulsant agent	Excitatory neuron activity inhibitor
Gliclazide	−85.24	1	Hypoglycemic agent	ATP sensitive potassium channel inhibitor
Ibutilide	−96.53	1	Antiarrhythmia agent	Potassium channel blocker
Imatinib	−99.33	1	Antineoplastic agent	Tyrosine kinase inhibitor
Iopamidol	−90.05	1	Radiographic contrast agent	X-ray contrast activity
Itraconazole	−95.47	5	Antiinfective agent (antifungal)	Fungal cytochrome P450 inhibitor (14-alpha demethylase inhibitors)
Ketoconazole	−87.97	2	Antiinfective agent (antifungal)	Fungal cytochrome P450 inhibitor (14-alpha demethylase inhibitors)
Levobunolol	−86.31	1	Sympatholytic agent	Beta-adrenergic receptor antagonist
Lonidamine	−89.97	1	Antineoplastic agent	Glucokinase inhibitor
Memantine	−97.66	1	Neuroprotective agent	*N*-methyl-d-aspartate glutamate receptor antagonist
Metolazone	−85.72	3	Antihypertensive agent (diuretic)	Sodium chloride symporter inhibitor
Nalidixic acid	−89.81	1	Antiinfective agent (antibacterial)	Bacterial topoisomerase II inhibitor
Niacin	−92.13	1	Antihypertensive agent (vasodilatator, hypolipidemic)	Lowering cholesterol
Nortriptyline	−92.84	2	Antidepressant agent	Adrenergic uptake inhibitor
Perindopril	−88.63	2	Antihypertensive agent	Angiotensin converting enzyme inhibitor
Reboxetine	−89.55	2	Antidepressant agent	Selective noradrenaline reuptake inhibitor
Rimantadine	−89.63	1	Antiinfective agent (antiviral)	Viral (influenza A) nucleic acid synthesis inhibitor
Ritonavir	−88.75	1	Antiinfective agent (antiviral)	Viral (HIV) protease inhibitor
Rufloxacin	−93.98	1	Antiinfective agent (antibacterial)	Bacterial topoisomerase II inhibitor
Spectinomycin	−93.99	1	Antiinfective agent (antibacterial)	Bacterial 30S ribosomal subunit inhibitor
Ticarcillin	−88.28	1	Antiinfective agent (antibacterial)	Inhibitor of bacterial cell wall synthesis
Tivozanib	−87.17	2	Antineoplastic agent	Vascular endothelial growth factor receptors inhibitor
Trimethadione	−96.08	1	Anticonvulsant agent	Inhibitor of voltage dependent T-type calcium channels
Triprolidine	−88.42	3	Antiallergic agent	Histamine receptor antagonist

## Data Availability

Not applicable.

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
