# Peer review of "Identification of 37 Heterogeneous Drug Candidates for Treatment of COVID-19 via a Rational Transcriptomics-Based Drug Repurposing Approach"

_pharmaceuticals, 2021, doi:10.3390/ph14020087_

Round 1

Reviewer 1 Report

Paper reports the study identify repurposable drug candidates with a potential to reverse transcriptomic alterations in the host cells infected by SARS-CoV-2. In detail authors identified 37 structurally heterogeneous drug candidates that revealed several biological processes as druggable pathways, including metabolic and biosynthetic processes, cellular developmental processes, immune response and signaling pathways, with steroid metabolic process being targeted by half of the drug candidates. The pipeline developed in this study integrates biological knowledge with rational study design and could be useful for future studies. Interestingly this approach support further investigations of some drugs currently in clinical trials (as itraconazole and imatinib) as useful agents for   COVID-19 treatment.

The paper is clear, well written and organized and obviously innovative.

In my opinion the major problem of this paper is related to cell lines selected for this study (A459 and Calu-3, page 4). This is a particular and non-obvious choise, as clearly cancer cell lines have a specific and characteristic transcriptomic profile, not strictly related to COVID-19 disease. The reason of this choise should be added, exploited and deeply justified by the authors.

In addition, some minor revisions are necessary:

  • Page 2, second paragraph: some brief phrases regarding drugs now in clinical trials for COVID-19 and corresponding references should be added (imatinib, Ruxolitinib, ibutinib and many others);
  • Regarding “inverted transcriptomic signatures”, firstly used in anticancer therapy, this reference should be added:

Koudijs, Kare, l K. M. et al Transcriptome Signature Reversion as a Method to Reposition Drugs Against Cancer for Precision Oncology The Cancer Journal: 3/4 2019 - Volume 25 - Issue 2 - p 116-120. doi: 10.1097/PPO.0000000000000370.

  • Tabel 2: MOA of amifostine is not clear; MOA of imatinib is not correct, being imatinib a Bcr-Ab inhibitor, consequently “Tyrosine kinase receptor inhibitor” should be changed with “Tyrosine kinase inhibitor”; MOA of Ticarcillina should be changed with “inhibitors of bacterial wall synthesis”.
  • Page 14, line 316: please delete “for identifying”.

For all these reasons, paper should be accepted after major revisions.

Reviewer 2 Report

The authors present a compelling study that employs development of a  rational computational pipeline to filter publicly available transcriptomic datasets from reported cell line studies with SARS-CoV-2 to identify drug candidates for repurposing drugs.

The overall study is well written and clear, given the conceptually complex design.  Figures help to guide design and filtering/selection criteria.

Question is how do the 2 currently FDA approved drugs fit into this analysis?  were they found as "hits"?  if not in final selection, at what level were they eliminated?   This is key to establish relevance of your pipeline for identifying relevant and plausible "hits".

Also, while the argument is given to narrow down cell line data sets to reduce noise,  one could argue that larger data set could strengthen assurance of identified drug set, especially since Fig 2 shows clear distinction (positive) vs. (negative) separation of all Sars-CoV-2 infected data sets to uninfected for all cell line sets.

A caution that filtering candidate drugs by criteria that they must have broad cell type activity is not necessarily sound as some key drugs may only work on epithelial cells and thus be "safer" (to avoid bystander toxicity) in cardiac or neural cells, for example.

Lastly, it could be argued that fold-changes in overexpression or suppression of genes is not biologically equal for all genes.  For enzyme activity for example a small change can have huge impact on cellular metabolism.  In contrast, protein expression of surface markers may have less sensitivity.  Should consider this in de-selection criteria.

Overall well done and important for the field.  Extensive analysis and Supplement tables and figures is important and helpful.   Suggest Fig S8 and S11 font on x-axis labels could be enlarged for easier reading (current size too small to read).

Round 2

Reviewer 1 Report

The major part of my suggestion has been followed by authors; this revised version of the manuscript could be accepted for publication.

This manuscript is a resubmission of an earlier submission. The following is a list of the peer review reports and author responses from that submission.